# Evolution of Occupational Safety and Health Disclosure Practices: Insights from 8 Years in Taiwan's Construction Industry

Chieh-Wen Chang [1], Tomohisa Nagata [2], Louise E. Anthony [3] and Ro-Ting Lin [1,*]

1 Department of Occupational Safety and Health, College of Public Health, China Medical University, Taichung 406040, Taiwan
2 Department of Occupational Health Practice and Management, Institute of Industrial Ecological Sciences, University of Occupational and Environmental Health, Kitakyushu 807-8555, Japan
3 International Master Program for Public Health, College of Public Health, China Medical University, Taichung 406040, Taiwan
* Correspondence: roting@mail.cmu.edu.tw; Tel.: +886-2-2205-3366

**Abstract:** The construction industry has been identified as a major contributor to occupational accidents that can lead to fatalities. As a result, this study aims to evaluate the effectiveness of new safety and health regulations and revised guidelines in improving safety and health disclosures and performance within the construction industry. We retrieved safety and health disclosure reports from 25 Taiwanese construction companies between 2013 and 2020 using the Market Observation Post System website. We analyzed the data using the Kaplan–Meier method to assess the timing of disclosures and differences between larger ($\geq$300 employees) and smaller (<300 employees) companies. We found that construction companies reported safety indicators more promptly than health indicators and that larger companies disclosed earlier compared to smaller ones. Only 45% of companies provide detailed reviews and preventative measures in their sustainability reports despite 64% disclosing occupational accidents. We found that from 2013 to 2020, more companies improved their occupational safety and health (OSH) reporting. This improvement coincided significantly with the adoption of international standards and Taiwan's government regulations. In summary, the study found that larger companies were more likely to disclose OSH data compared to smaller ones. This suggests that company size and available resources could have an impact on reporting practices. While some progress was made, companies still struggle to provide detailed reports on major accidents, balancing transparency with competitiveness.

**Keywords:** construction; disclosure; occupational safety; regulations; Taiwan

## 1. Introduction

The construction industry is recognized for its high rate of occupational accidents, surpassing those in other sectors globally [1,2]. Meta-analyses studies from Canada, Germany, Spain, the United Kingdom, and the United States have underscored the positive impact of occupational safety and health (OSH) regulations in mitigating these accidents [3]. Conversely, evidence from South Korea emphasizes the adverse outcomes associated with deregulation, including the underreporting and concealment of work-related accidents [4]. While regulations serve as critical organizational interventions providing a comprehensive framework for companies' OSH initiatives [5], relying solely on them may not lead to a substantial reduction in injuries. Their effectiveness is bolstered by the proactive implementation of practical measures undertaken by companies [5].

In 2013, Taiwan made amendments to the Occupational Safety and Health Act [6,7]. This action extended legal protections to all workers, establishing a system for the management of machinery, equipment, and chemicals from the source, and strengthening

the prevention system for occupational diseases. Additionally, the amendment includes provisions for maternal protection and employment equity, enhances safety supervision and penalties for high-risk industries, promotes workplace safety and health culture, and stipulates that workers can evacuate themselves under certain circumstances to avoid danger [8,9]. The revisions incentivized companies to implement self-management mechanisms overseen by the Financial Supervisory Commission and non-compliance would result in legal penalties based on the severity of the case [10,11]. The self-management approach emphasizes the significance of formulating a comprehensive policy, implementing effective measures, assessing performance, and implementing necessary improvements, thereby establishing a foundation for a safer work environment [12]. This approach closely aligns with most current Occupational Health and Safety Management Systems (OHSMS), which integrate the Plan–Do–Check–Act cycle to ensure continuous improvement and optimization of occupational health and safety practices [13]. Studies, including one conducted in the Korean construction sector, have demonstrated a significant reduction in occupational accidents after the implementation of OHSMS [14].

In recent years, companies have frequently expressed their commitment to employee safety through corporate social responsibility or sustainability reports (hereafter referred to as sustainability reports), which elaborate on their safety policies and actions [15]. In Taiwan, by the end of 2014, both the Taiwan Stock Exchange and Taipei Exchange announced Rules Governing the Preparation and Filing of Corporate Social Responsibility Reports on November 26, 2014, and December 4, 2014, respectively [16–19]. These rules stipulated that companies must prepare their sustainability reports in accordance with the Global Reporting Initiative (GRI) [20]. In Japan, the Tokyo Stock Exchange's "Corporate Governance Code" encourages companies to publish sustainability reports, often following GRI guidelines [21]. Starting in 2025, South Korea requires listed companies with assets exceeding KRW 2 trillion to publish sustainability reports, expanding this requirement to all listed companies by 2030 [22]. The Korea Exchange has also released ESG Disclosure Guidelines to provide advice on preparing sustainability reports [23]. In Taiwan, from 2025, listed companies with paid-in capital of less than TWD 2 billion will be required to prepare sustainability reports [24]. Overall, Japan, Taiwan, and South Korea are all promoting the disclosure of sustainability reports by listed companies, reflecting the efforts of Asian countries to enhance corporate transparency and responsible management. This also illustrates how global sustainability trends are influencing regional policies. The advantage of cultivating a positive corporate image regarding occupational safety and health extends to enhance productivity, increase investor appeal, and improve financial outcomes. Transparency and the quality of safety and health information disclosures play a pivotal role in fostering trust among workers and stakeholders [25].

Given the elevated rate of occupational accidents in industries such as construction, this study examines the inclination of companies to disclose their safety performance in response to regulatory changes. Both the Taiwan Stock Exchange Corporation and Taipei Exchange have mandated sustainability reporting [16–19], which has positively impacted corporate reputation and performance [26–28]. This research aims to analyze the trends in the disclosure of safety and health actions and performance within Taiwan's construction industry from 2013 to 2020. It seeks to understand how the industry's commitment to OSH has evolved, potentially influenced by regulatory changes after 2013, and how this evolution reflects in the transparency and communication of OSH initiatives.

## 2. Materials and Methods

### 2.1. Study Design and Study Period

This longitudinal study explores the disclosure of corporate safety and health actions and performance disclosures by Taiwanese construction companies from 2013 to 2020. The primary data source consists of reports on these actions and performances, typically published with a one-year delay (i.e., from 2014 to 2021). To ensure clarity and relevance,

our study period referenced the actual years of actions and performance (2013–2020), rather than the publication years of the reports.

### 2.2. Study Samples and Settings

Our study focused on construction companies listed on the Taiwan Stock Exchange Corporation and Taipei Exchange. Companies that are listed on these exchanges are required by law to publish sustainability reports [16–19]. We selected companies based on their sustainability reports available on the Market Observation Post System and company websites throughout the study period [29]. In total, we obtained 128 reports from 25 companies for variable extraction.

### 2.3. Variables

To evaluate corporate safety and health actions and performance, data were collected on four aspects, encompassing a total of 15 indicators (Table 1), aligned with GRI [20]. Specifically, these four aspects are: (1) recognition of occupational safety and health (hereafter referred to as 'Recognitions'), comprising four indicators; (2) occupational safety and health goals (hereafter referred to as 'Goals'), consisting of three indicators; (3) occupational safety and health measures (hereafter referred to as 'Measures'), incorporating five indicators; and (4) occupational safety and health outcomes (hereafter referred to as 'Outcomes'), consisting of three indicators. In this study, the criterion for categorizing company size was based on the number of employees derived from the companies' annual reports for the year 2013.

According to the GRI 403 standards, reporting organizations are advised to disclose information on work-related injuries, including details such as the total number and rate of work-related fatalities, the number and rate of severe work-related injuries (excluding fatalities), the number and rate of recordable work-related injuries, the primary types of work-related injuries, and the total number of hours worked [30]. However, in practice, companies are not obligated to disclose information related to occupational accidents. However, to verify this aspect, we checked the Occupational Safety and Health Administration of Taiwan's public website for major occupational accidents in the construction industry [31]. This involved examining the year and the number of major occupational accidents reported during the study period. Subsequently, we reviewed the company's sustainability reports related to these incidents to determine if the companies disclosed the following information in the same year: disclosure of occupational accidents, disclosure of reviews on the causes of occupational accidents, and disclosure of preventive measures for occupational accidents.

**Table 1.** Occupational safety and health action and performance indicators.

| Aspects and Indicators | Definition | Reference |
|---|---|---|
| 1. Recognitions | | |
| 1.1 Occupational safety and health chapter | The occupational safety and health-related chapters are mentioned in the table of contents or Appendix A, and there are indeed paragraphs within the report discussing occupational safety and health. | [32] |
| 1.2 Material topics | The report mentions the aggregation of stakeholder engagement levels for conducting a significant analysis of major issues. | [32] |
| 1.3 Occupational safety as a material topic | List occupational safety or occupational safety and health as a material topic. | [32] |
| 1.4 Occupational health as a material topic | List occupational health or occupational safety and health as a material topic. | [32] |

**Table 1.** *Cont.*

| Aspects and Indicators | Definition | Reference |
|---|---|---|
| 2. Goals | | |
| 2.1 Occupational safety goals | The report mentions quantitative goals related to occupational safety. | [32] |
| 2.2 Occupational health goals | The report mentions quantitative goals related to occupational health. | [32] |
| 2.3 ESG/SDGs | The report mentions the alignment with ESG principles and/or SDGs. | [32] |
| 3. Measures | | |
| 3.1 Occupational safety education | The report mentions that the company offers education or training regarding occupational safety and accident prevention. | [30] |
| 3.2 Occupational health education | The report mentions that the company provides education or training regarding occupational health and health promotion. | [30] |
| 3.3 Occupational safety and health committee | The report mentions that the company has an occupational safety and health committee or clearly states that the safety and health committee is comprised of an existing labor-management meeting. | [30] |
| 3.4 Occupational health and safety management system | The report mentions that the company has obtained occupational safety and health-related certifications (e.g., ISO 45001, ISO 18001, TOSHMS, or CNS 15506). | [30] |
| 3.5 Mental health support | The report mentions that the company has an EAP and offers psychological counseling services to employees. | [30] |
| 4. Outcomes | | |
| 4.1 Occupational accident-related outcomes | The report contains information such as occupational accidents, work-related injuries, and cases of occupational diseases. | [30] |
| 4.2 Quantitative data on occupational injury and illness | The report shows quantitative data on disabilities, lost workdays, and the number of sick leave instances due to work-related injuries and illnesses. | [30] |
| 4.3 Near miss | The report provides information on near misses, such as their frequency, how they are managed, and the methods used for improvement. | [30] |

Note: CNS 15506 = Taiwan Occupational Safety and Health Management System Verification Specification 15506; EAP = Employee Assistance Program; ESG = Environmental, Social, and Governance; ISO 45001 = ISO 45001 Occupational Health and Safety Management Systems; ISO 18001 = Occupational Health and Safety Assessment Series 18001; SDGs = Sustainable Development Goals; TOSHMS = Taiwan Occupational Safety and Health Management System.

*2.4. Data Sources, Management, and Bias*

We conducted a thorough data collection process involving the acquisition of sustainability reports from the Market Observation Post System and company websites [29]. Subsequently, an examination of these reports was undertaken to evaluate the presence of 15 key indicators, pertaining to safety and health, categorized across four aspects. Each indicator was coded as disclosed or not by the study companies. The data on the number of employees were also extracted from the annual reports [33], while information on significant occupational accidents was sourced from the Major Occupational Accident Disclosure website of Taiwan [31].

To ensure the reliability and accuracy of data collection, a systematic approach was adopted. The primary researcher, alongside two additional trained collectors, conducted

a rigorous examination of the texts, and coded data based on pre-defined criteria. Before conducting data collection, it is crucial to ensure that all data collectors possess the necessary professional knowledge and experience. In this study, both the primary researcher and the data collectors have experience in collecting and reviewing at least 50 reports. This extensive background enables them to accurately identify key information and relevant data, thereby ensuring that the results evaluated are of good quality and reliability. Any disagreements were resolved by a senior researcher to ensure data integrity. The internal consistency of indicators was assessed using Cronbach's alpha [34,35], with a coefficient of 0.83 indicating high consistency [36], thereby affirming the reliability of our indicators in measuring a common construct [37]. Furthermore, inter-rater agreement, as evaluated through Cohen's kappa, demonstrated strong consistency across all indicators, with values ranging from moderate (0.52) to almost perfect agreement (1.00) (refer to Table A1) [38].

### 2.5. Statistical Analysis

We quantified the disclosures of company's individual indicators by computing their frequency and percentage. To evaluate the timing of safety and health disclosures across 15 specific indicators, we employed the Kaplan–Meier method. We conducted subgroup analyses to identify differences based on company size, applying the logrank test to contrast disclosure timings between these groups. In Taiwan, company size was determined by the number of employees reported in the annual report for the initial year (i.e., 2013) [33], according to the Ministry of Labor, with companies of 300 or above 300 employees classified as large and those below 300 as small [39]. All statistical analyses were analyzed using SAS version 9.4 (SAS Institute Inc., Cary, NC, USA), with a Type I error threshold set at 0.05.

### 2.6. Ethical Statements

The data for this study were collected exclusively from publicly available corporate reports and websites. Therefore, the study did not involve any human subjects, and no personal or sensitive information was obtained. Thus, ethical approval was not required for this research. The use of publicly accessible data adheres to principles of data privacy and confidentiality.

## 3. Results

Table 2 shows the number and percentage of companies that published sustainability reports and those that reported on their occupational safety and health actions and performance between 2013 to 2020. The number of companies that published these reports increased from 2 companies in 2013 to 24 by 2020. Table 2 also demonstrates increases in the number of companies disclosing all indicators at varying levels. Notably, all 'Recognitions' indicators were disclosed by approximately 50% of companies in 2013, escalating to 83% or more by 2020. 'Goals' related to occupational safety and ESG/SDGs achieved a disclosure rate of 50% or higher by 2014 and 2017, respectively, with over 75% by 2020, while disclosures of occupational health goals remained below 5%. Regarding the 'Measures' aspect, disclosures on safety and health education reached 50% by 2013 and 2020, respectively, with other measures under 40% between 2013 and 2020. Concerning the 'Outcomes' aspect, accident-related outcomes and quantitative data were disclosed by 50% of companies since 2013 and 2014, respectively, but near-miss disclosures remained below 10% as of 2020.

Table 3 displays the average duration of disclosure for 15 indicators ranging from 3.60 years for the 'occupational safety and health chapter' to 8.00 years for 'occupational health goals.' The 8.00 year duration in the survival function may suggest a lack of disclosures throughout the observation period or a singular disclosure event in the final year. This could signify that either no companies disclosed the specified information until the end, or one company made its disclosure in the eighth year. When comparing safety and health disclosures, safety indicators were disclosed more promptly than health indicators: material topics were disclosed in 4.76 years (safety) versus 4.80 years (health), goals in 5.24 years (safety) versus 8.00 years (health), and education topics in 4.32 years (safety) versus 6.20 years (health).

Analysis by company size shows that larger companies tend to disclose most of the indicators more promptly than smaller ones (Table 3), with significant differences in disclosures of: occupational safety goals (4.00 years (large) vs. 6.21 years (small), $p = 0.046$), ESG/SDGs (5.00 years (large) vs. 6.57 years (small), $p = 0.008$), occupational safety education (3.36 years (large) vs. 5.07 years (small), $p = 0.027$), occupational health education (5.36 years (large) vs. 6.86 years (small), $p = 0.035$), OHSMS system (4.09 years (large) vs. 7.86 years (small), $p = 0.017$), occupational accident-related outcomes (3.36 years (large) vs. 5.14 years (small), $p = 0.028$), and quantitative data on occupational injury and illness (3.82 years (large) vs. 5.86 years (small), $p = 0.007$). Detailed survival function plots for each indicator are provided in Figure A1.

**Table 2.** Number and percentage of companies reporting 15 indicators from 2013 to 2020.

| Aspects and Indicators | 2013 (N = 2) | 2014 (N = 10) | 2015 (N = 11) | 2016 (N = 19) | 2017 (N = 20) | 2018 (N = 21) | 2019 (N = 21) | 2020 (N = 24) |
|---|---|---|---|---|---|---|---|---|
| 1. Recognitions | | | | | | | | |
| 1.1 Occupational safety and health chapter | 2 (100%) | 9 (90%) | 10 (91%) | 18 (95%) | 18 (90%) | 19 (90%) | 20 (95%) | 23 (96%) |
| 1.2 Material topics | 1 (50%) | 8 (80%) | 10 (91%) | 18 (95%) | 19 (95%) | 20 (95%) | 20 (95%) | 23 (96%) |
| 1.3 Occupational safety as a material topic | 1 (50%) | 6 (60%) | 4 (36%) | 13 (68%) | 14 (70%) | 15 (71%) | 15 (71%) | 21 (88%) |
| 1.4 Occupational health as a material topic | 1 (50%) | 6 (60%) | 5 (45%) | 12 (63%) | 12 (60%) | 14 (67%) | 15 (71%) | 20 (83%) |
| 2. Goals | | | | | | | | |
| 2.1 Occupational safety goals | 0 (0%) | 5 (50%) | 6 (55%) | 11 (58%) | 12 (60%) | 13 (62%) | 15 (71%) | 18 (75%) |
| 2.2 Occupational health goals | 0 (0%) | 0 (0%) | 0 (0%) | 0 (0%) | 0 (0%) | 0 (%) | 0 (0%) | 1 (4%) |
| 2.3 ESG/SDGs | 0 (0%) | 1 (10%) | 2 (18%) | 7 (37%) | 12 (60%) | 15 (71%) | 16 (76%) | 20 (83%) |
| 3. Measures | | | | | | | | |
| 3.1 Occupational safety education | 1 (50%) | 7 (70%) | 7 (64%) | 14 (74%) | 17 (85%) | 18 (86%) | 18 (86%) | 21 (88%) |
| 3.2 Occupational health education | 0 (0%) | 3 (30%) | 3 (29%) | 7 (37%) | 7 (35%) | 8 (38%) | 7 (33%) | 13 (54%) |
| 3.3 Occupational safety and health committee | 0 (0%) | 3 (30%) | 3 (27%) | 7 (37%) | 7 (35%) | 8 (38%) | 8 (38%) | 9 (38%) |
| 3.4 Occupational health and safety management system | 0 (0%) | 2 (20%) | 3 (27%) | 5 (26%) | 6 (30%) | 7 (33%) | 7 (33%) | 8 (33%) |
| 3.5 Mental health support | 0 (0%) | 0 (0%) | 1 (9%) | 0 (0%) | 3 (15%) | 3 (14%) | 2 (10%) | 4 (17%) |
| 4. Outcomes | | | | | | | | |
| 4.1 Occupational accident-related outcomes | 1 (50%) | 8 (80%) | 8 (73%) | 15 (79%) | 15 (75%) | 17 (81%) | 18 (86%) | 21 (88%) |
| 4.2 Quantitative data on occupational injury and illness | 0 (0%) | 7 (70%) | 6 (55%) | 12 (63%) | 13 (65%) | 14 (67%) | 17 (81%) | 14 (58%) |
| 4.3 Near miss | 0 (0%) | 0 (0%) | 0 (0%) | 0 (0%) | 1 (5%) | 1 (5%) | 1 (5%) | 2 (8%) |

Note: ESG = Environmental, Social, and Governance; N = number of companies; SDGs = Sustainable Development Goals.

**Table 3.** Mean and standard errors of years to disclosure for occupational safety and health actions and performance for the overall sample and groups.

| Aspects and Indicators | Total | Company Size | | |
|---|---|---|---|---|
| | | Small | Large | *p* Value |
| 1. Recognitions | | | | |
| 1.1 Occupational safety and health chapter | 3.60 (0.42) | 4.00 (0.69) | 3.09 (0.34) | 0.230 |
| 1.2 Material topics | 3.92 (0.43) | 4.50 (0.70) | 3.18 (0.33) | 0.088 |
| 1.3 Occupational safety as a material topic | 4.76 (0.49) | 5.21 (0.75) | 4.18 (0.58) | 0.108 |
| 1.4 Occupational health as a material topic | 4.80 (0.50) | 5.21 (0.75) | 4.27 (0.62) | 0.125 |
| 2. Goals | | | | |
| 2.1 Occupational safety goals | 5.24 (0.50) | 6.21 (0.61) | 4.00 (0.68) | 0.046 |
| 2.2 Occupational health goals | 8.00 [a] (*) | 8.00 [b] (*) | 8.00 [a] (*) | 0.259 |
| 2.3 ESG/SDGs | 5.88 (0.38) | 6.57 (0.55) | 5.00 (0.40) | 0.008 |

**Table 3.** *Cont.*

| Aspects and Indicators | Total | Company Size | | |
|---|---|---|---|---|
| | | Small | Large | *p* Value |
| 3. Measures | | | | |
| 3.1 Occupational safety education | 4.32 (0.56) | 5.07 (0.72) | 3.36 (0.36) | 0.027 |
| 3.2 Occupational health education | 6.20 (0.46) | 6.86 (0.65) | 5.36 (0.60) | 0.035 |
| 3.3 Occupational safety and health committee | 6.20 (0.47) | 7.00 (0.51) | 4.54 (0.54) | 0.105 |
| 3.4 Occupational health and safety management system | 6.80 (0.44) | 7.86 (0.19) | 4.09 (0.38) | 0.017 |
| 3.5 Mental health support | 7.20 (0.32) | 7.64 (0.49) | 5.55 (0.22) | 0.093 |
| 4. Outcomes | | | | |
| 4.1 Occupational accident-related outcomes | 4.36 (0.48) | 5.14 (0.73) | 3.36 (0.43) | 0.028 |
| 4.2 Quantitative data on occupational injury and illness | 4.96 (0.48) | 5.86 (0.68) | 3.82 (0.52) | 0.007 |
| 4.3 Near miss | 7.88 (0.17) | 8.00 [a] (*) | 5.00 (*) | 0.838 |

Note: ESG = Environmental, Social, and Governance; SDGs = Sustainable Development Goals. * The value represents companies that do not have a standard error value for statistical analysis. [a] During the study period with company disclosures, the average disclosure time was 8 years. [b] During the study period, there were no company disclosures, hence the average survival time was 8 years.

After conducting a search of the publicly available information on major occupational accidents provided by the Occupational Safety and Health Administration of Taiwan, we observed that the earliest published data were from 2017. During the period spanning 2013 to 2020, our study identified a total of 15 major occupational accidents occurring within the study companies. Table 4 shows that among the 25 sample companies, 11 had experienced major occupational accidents, totaling 15 incidents. Of these 11 companies, 4 were identified as public institutions and 7 as business owners. Furthermore, 7 companies disclosed the incidents of major occupational accidents, and 5 disclosed their reviews and preventive measures. However, no company provided a detailed description of the occurrence process of the major occupational accidents.

**Table 4.** Extent of disclosure of major occupational accident incidents.

| Company ID | Year | Affiliation Status | Number of Major Occupational Accidents | Disclosure of Major Occupational Accidents | Detailed Explanation of the Occurrence of Major Occupational Accidents | Disclosure of Reviews and Preventive Measures |
|---|---|---|---|---|---|---|
| 1 | 2017 | Business owner | 1 | No | No | No |
| 2 | 2017 | Business owner | 1 | No | No | No |
| 3 | 2018 | Business owner | 1 | No | No | No |
| 4 | 2018 | Public institutions | 1 | Yes | No | Yes |
| 5 | 2018 | Business owner | 1 | Yes | No | Yes |
| 6 | 2018 | Public institutions | 1 | Yes | No | No |
| 7 | 2019 | Public institutions | 4 | Yes | No | No |
| 8 | 2019 | Business owner | 1 | Yes | No | Yes |
| 9 | 2019 | Public institutions | 1 | Yes | No | Yes |
| 10 | 2019 | Business owner | 1 | No | No | No |
| 11 | 2019 | Business owner | 1 | No | No | No |

## 4. Discussion

This study highlights the evolution of OSH disclosure practices in Taiwan's construction industry. In 2013, amendments to the Occupational Safety and Health Act led to a shift towards improved sustainability reporting and adaptation of OSH measures. This change indicates a significant advancement in the industry's transparency and commitment to safety and health at work. Our findings revealed that there were differences in disclosure times across indicators and company sizes, especially concerning 'health' indi-

cators. Smaller companies took longer to disclose highlighting the importance of robust organizational resources and effective OSH management to accelerated reporting practices.

Notably, construction companies have made significant improvements in their safety and health indicators from 2013 to 2020, resulting from regulatory changes made by the government. These regulatory changes were the result of amendments to the Construction Industry Act in 2011, the Occupational Safety and Health Act in 2013, and the Construction Safety and Health Facility Standard in 2014 [7,40,41]. As a result of these changes, companies have enhanced their focus on safety and health, leading to increased disclosures of related actions and performances. The rise in safety and health disclosures since 2013 underscores the positive influence of these regulatory changes on promoting transparency within Taiwan's construction industry. Across the indicators, our analysis revealed a spectrum from 3.60 years for the 'occupational safety and health chapter' to 8.00 years for 'occupational health goals.' This significant change was attributed to the introduction of the GRI standards 103 and 403 in 2016 [30,32]. GRI 103 is tailored to material topics whereas GRI 403 focuses on occupational safety and health [30,32]. This indicates that companies are gradually aligning with these external changes, particularly noting faster disclosures of policies under GRI 103 compared to OSH details under GRI 403, reflecting the progressive integration of these guidelines into corporate reporting practices. The materiality principle is highlighted through the initial focus on the 'occupational safety and health chapter' and 'material topics' [42]. However, the significance of occupational health goals received comparatively less emphasis, possibly due to the fact that occupational safety is more visibly apparent. Although occupational health is equally important, it has historically been given less immediate attention and has played a supportive role in company policies and the Occupational Safety and Health Act [43].

Larger companies tend to disclose OSH indicators faster than smaller ones, with 47% of indicators being disclosed significantly quicker in our analysis. This is partly due to regulatory mandates related to company size and risk exposure [39]. Under Taiwan's Occupational Safety and Health Management Regulations, all construction businesses must designate a safety and health supervisor [39]. Companies with over 300 employees must appoint a safety (or health) manager and two safety and health officers. Smaller companies, on the other hand, are only required to have one safety and health officer and are not mandated to appoint a manager. This regulatory framework significantly affects the way companies report their occupational safety and health indicators. Moreover, differences in disclosure practices across company sizes may be attributed to heightened stakeholder expectations and the superior financial and infrastructural resources of larger companies. These resources facilitate a stronger dedication to corporate social responsibility [44] and the capacity to implement expensive health promotion initiatives [45,46]. It is evident from the measures taken to manage occupational safety and health that larger companies are primarily driven to adopt an OHSMS to comply with legal requirements, support employee welfare and job satisfaction, showcase social responsibility, and improve the company's internal and external image [47]. However, smaller companies often encounter challenges such as limited resources and infrastructure, which can impede their ability to promptly disclose their health and safety initiatives. On the other hand, larger companies tend to have more resources and may engage in more visible initiatives, although all companies adhere to regulatory standards. However, it is important to note that, regardless of size, all companies are committed to taking actions that promote safety and health in their workplaces. In the construction industry, companies may face certain implications due to government and internal standards. This includes the need to disclose information as per regulatory changes, shorter disclosure times as per sub-regulations after the revision of GRI 403 in 2016, and the requirement of assistance from the government for smaller companies to disclose information.

This study focused on major occupational accidents in the construction industry, which occur more frequently compared to other sectors. Despite their significance, companies tend to disclose a relatively low proportion of these accidents in their sustainability reports.

Moreover, companies are often reluctant to provide detailed information about incidents and the measures taken to review and improve the situation. This reluctance can be attributed to concerns about competitive threat, even though disclosing such information can help establish a positive corporate image and build trust with stakeholders. These findings are consistent with previous research that highlights the benefits of disclosing workplace information [48].

*Limitations*

Our study has a few limitations. First, we only found 25 listed construction companies that published a total of 128 sustainability reports from 2013 to 2020. According to statistics from the Ministry of Economic Affairs, as of December 2020, there were 108,597 registered construction companies in Taiwan [49]. However, there is no legal requirement for non-listed companies to publish sustainability reports. Since listed companies are generally largescale, they often subcontract many smaller, non-listed construction companies, so some non-listed companies might be included in the sustainability reports issued by these listed companies [16–19]. Hence, we suggest conducting further research to examine the strategies employed by larger companies to manage their subcontractors.

Second, for statistical analysis, we evaluated reports using indicators to obtain the company's disclosure status on these metrics. In our study, although the sample consists of 25 listed companies, as this is a longitudinal study, the actual sample includes 128 reports from 2013 to 2020. However, we believe that this sample size is still insufficient to provide the necessary generalizability and statistical power. We recommend conducting long-term research in the future to collect substantial data, thus avoiding issues with insufficient sample sizes.

It is important to note that before 2011, construction companies did not often publish sustainability reports, making it difficult to retrieve data. These limitations emphasize the need for broader data collection across the industry and extended timeframes to improve the study's applicability and accuracy. Moreover, not all sample companies published sustainability reports in 2013. As a result, employee numbers obtained from annual reports were used to group companies by size. During the data collection process, we observed discrepancies in the number of employees disclosed in sustainability reports and annual reports. This variation is probably due to the different purposes and scopes of these two types of reports. Annual reports, which are prepared according to financial reporting standards, cover all types of employees to provide a comprehensive view of the company's operational status. In contrast, sustainability reports focus more on showcasing the company's performance in environmental, social, and governance aspects and tend to highlight the core employees involved in these activities.

## 5. Conclusions

This research highlights the evolution in OSH disclosure within Taiwan's construction industry from 2013 to 2020, propelled by government regulations and GRI standards that have heightened awareness and reporting of safety and health information. The identified disparities in disclosure timing across different company sizes suggest a need for targeted governmental support or incentives, especially for smaller enterprises and those confronting substantial regulatory or risk-based reporting requirements. To promote safer working environments, we recommend that the government offer incentives to small-scale companies to encourage them to produce disclosure reports on occupational safety and health. This will not only help improve the overall safety and health standards of companies but also ensure the well-being of employees. Overall, while significant progress has been made in the disclosure of occupational safety and health in Taiwan's construction industry, there remains a need to enhance the disclosure and reporting of major occupational accidents and related improvement measures. Specifically, there is a call for further support and guidance from the government and relevant institutions to

improve the quality and timeliness of disclosures by smaller businesses and companies facing significant challenges.

**Author Contributions:** Conceptualization, R.-T.L.; data curation, C.-W.C.; formal analysis, C.-W.C.; funding acquisition, R.-T.L.; methodology, R.-T.L.; supervision, R.-T.L.; validation, C.-W.C.; writing—original draft, C.-W.C. and R.-T.L.; writing—review and editing, C.-W.C., T.N., L.E.A. and R.-T.L. All authors have read and agreed to the published version of the manuscript.

**Funding:** This research was funded by the National Science and Technology Council, Taiwan under grant numbers 111-2813-C-039-187-B and 111-2314-B-039-020-MY2 and China Medical University, Taiwan under grant numbers CMU111-SR-140, CMU112-S-26, and CMU112-MF-76. The funding source had no role in the study design, data collection, data analysis, data interpretation, writing of the manuscript, or decision to submit the paper for publication.

**Institutional Review Board Statement:** Not applicable.

**Informed Consent Statement:** Not applicable.

**Data Availability Statement:** The data that support the findings of this study are available from the corresponding author upon reasonable request.

**Acknowledgments:** We acknowledge grant support from the National Science and Technology Council, Taiwan, and China Medical University, Taiwan.

**Conflicts of Interest:** The authors declare they have no known competing financial interests or personal relationships that could have appeared to influence the work reported in this paper.

## Appendix A

**Table A1.** The kappa value of the correct data comparison of the three data collectors.

| Aspects and Indicators | Primary Researcher (Data Collector 1) | Data Collector 2 | Data Collector 3 |
|---|---|---|---|
| 1. Recognitions | | | |
| 1.1 Occupational safety and health chapter | 0.98 | 0.94 | 0.81 |
| 1.2 Material topics | 0.98 | 1.00 | 0.91 |
| 1.3 Occupational safety as a material topic | 0.94 | 0.95 | 0.73 |
| 1.4 Occupational health as a material topic | 0.84 | 0.92 | 0.73 |
| 2. Goals | | | |
| 2.1 Occupational safety goals | 1.00 | 0.80 | 0.77 |
| 2.2 Occupational health goals | 1.00 | 0.97 | 0.94 |
| 2.3 ESG/SDGs | 0.97 | 0.86 | 0.52 |
| 3. Measures | | | |
| 3.1 Occupational safety education | 0.83 | 0.73 | 0.75 |
| 3.2 Occupational health education | 0.91 | 0.78 | 0.80 |
| 3.3 Occupational safety and health committee | 0.91 | 0.91 | 0.78 |
| 3.4 Occupational health and safety management system | 0.97 | 0.94 | 0.89 |
| 3.5 Mental health support | 0.92 | 0.88 | 0.88 |
| 4. Outcomes | | | |
| 4.1 Occupational accident-related outcomes | 0.94 | 0.84 | 0.78 |
| 4.2 Quantitative data on occupational injury and illness | 0.94 | 0.81 | 0.81 |
| 4.3 Near miss | 1.00 | 0.93 | 0.97 |

Note: ESG = Environmental, Social, and Governance; SDGs = Sustainable Development Goals.



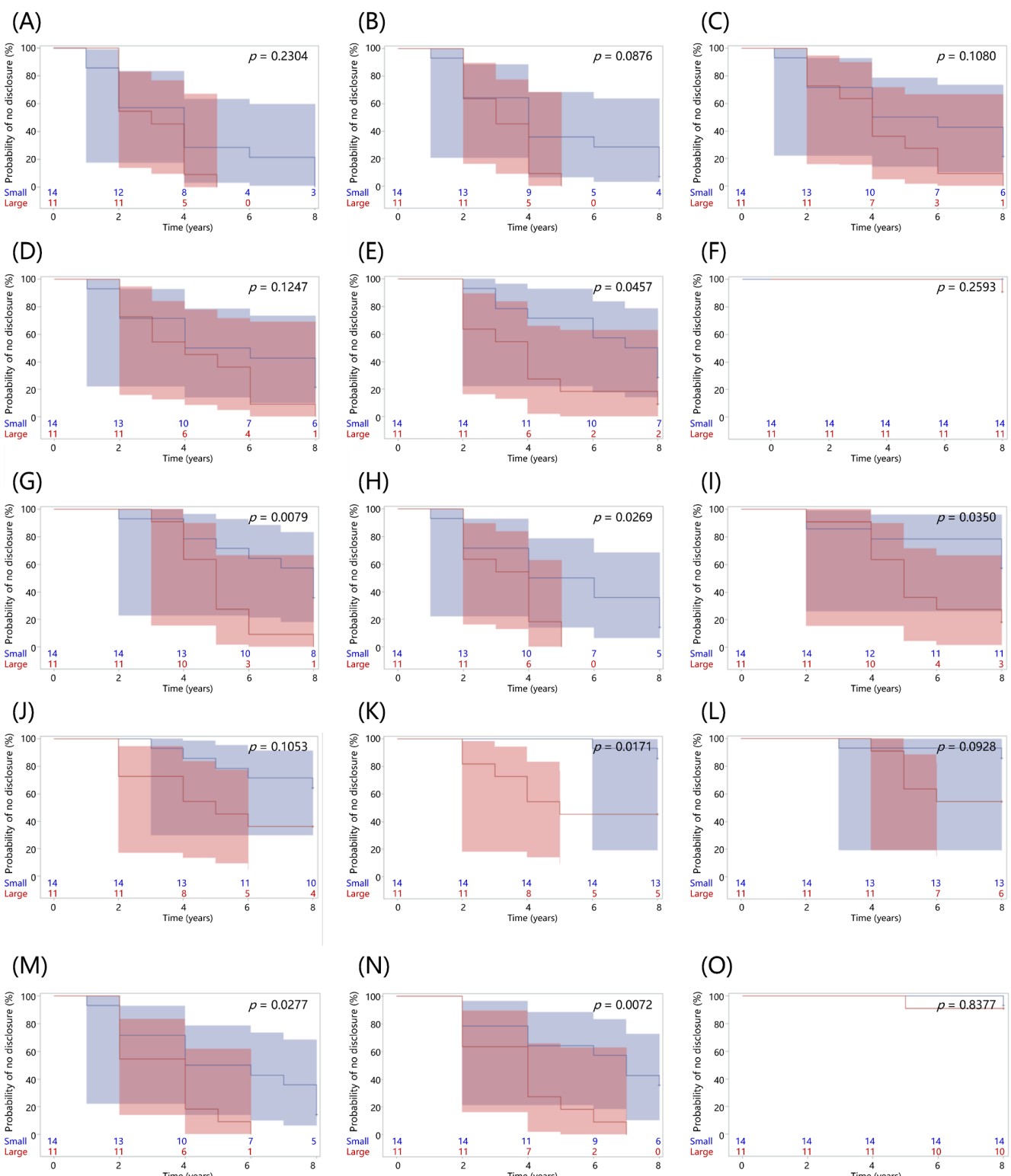

**Figure A1.** Survival function plot of the disclosure indicator grouping companies by average number of employees. (**A**) The survival function graph of disclosing "occupational safety and health chapter" by companies, (**B**) the survival function graph of disclosing "material topics" by companies, (**C**) the survival function graph of disclosing "occupational safety as a material topic" by companies, (**D**) the survival function graph of disclosing "occupational health as a material topic" by companies, (**E**) the survival function graph of disclosing "occupational safety goals" by companies, (**F**) the survival function graph of disclosing "occupational health goals" by companies, (**G**) the survival function graph of disclosing



"ESG/SDGs" by companies, (**H**) the survival function graph of disclosing "occupational safety education" by companies, (**I**) the survival function graph of disclosing "occupational health education" by companies, (**J**) the survival function graph of disclosing "occupational safety and health committee" by companies, (**K**) the survival function graph of disclosing "occupational health and safety management system" by companies, (**L**) the survival function graph of disclosing "mental health support" by companies, (**M**) the survival function graph of disclosing "occupational accident-related outcomes" by companies, (**N**) the survival function graph of disclosing "quantitative data on occupational injury and system" by companies, and (**O**) the survival function graph of disclosing "near miss" by companies. Note: (**F,K,O**) represent smaller companies with limited information on disclosure of indicators. Therefore, the survival time cannot be determined. In each figure, survival curves are presented for smaller companies (blue curve) and larger companies (red curve), accompanied by 95% Hall-Wellner confidence bands around each curve. These bands indicate the uncertainty associated with the survival estimates for each group.

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
