# Peer review of "Evolution of Occupational Safety and Health Disclosure Practices: Insights from 8 Years in Taiwan’s Construction Industry"

_safety_

Round 1
Reviewer 1 Report
Comments and Suggestions for Authors
The study highlights improvements in OSH disclosure in the Taiwanese construction industry, driven by government regulations and GRI standards. There is a need to increase the disclosure of major work accidents and associated remedial measures, particularly for small businesses and companies facing several challenges, through additional support and guidance.
Overall, this new longitudinal study shows how these regulatory changes affect the Taiwan Occupational Safety and Health Code and the adoption of the GRI standards has improved productivity in internal safety and health (OSH) disclosures in sustainability reports issued by construction companies, especially large projects. Using the Kaplan-Meier method to analyze disclosure time across 15 OSH indicators, provides useful insights into the impact of changing regulations on corporate OSH transparency and reporting gaps that should be addressed, emphasizing the study’s potential to identify policies that increase disclosure behaviors.
But there are still some errors and omissions in the article that need to be corrected and supplemented:
The table heading should show in the second page for Table 2, 3 and Table 4.
The author's discussion of the results in the "Results" section needs to be clearer and supported by more evidence. This section should provide additional explanation and context regarding the implications and significance of the results presented.
The "Limitations" section should propose solutions to address the identified problems.
In the "Conclusions" section, the author should offer perspectives on the future direction of the research.
The manuscript presents a logical and coherent study, but some mistakes need to be corrected, as mentioned above.
Comments on the Quality of English LanguageThe manuscript has good English presentation.
Author Response
We sincerely appreciate your time and effort reviewing our manuscript and providing valuable feedback. We have diligently addressed all suggestions and incorporated revisions accordingly. We have implemented all revisions using the track change function, and a detailed point-by-point response to each comment as shown in the attachment is provided for your review. We believe these enhancements have significantly improved the quality and depth of our manuscript. We sincerely thank you once again for your thorough review and constructive suggestions.

Reviewer 2 Report
Comments and Suggestions for Authors
This study provided an interesting perspective on the health and safety aspect of the local market, however the following aspects require additional discussion or explanation.
1.Abstract requires improvement. In a single paragraph the aims, scope and obtained results should be clearly described.
2.Please explain what a small and large company means (it would be best to specify the number of employees).
3.The presented text does not indicate whether reports in all construction companies are obligatory in accordance with applicable regulations or whether they do not have to be presented?
4. Please also explain who has access to reports on occupational health and safety?
5. Please describe in detail the extent to which the law has changed since 2013 in the aspect of occupational health and safety?
6.There is no literature review chapter that should discuss how other countries with a similar political structure as Taiwan report occupational health and safety results and what legal provisions they have regarding this issue. please present on a comparative basis.
Author Response

(The authors gave the same response as above.)

Reviewer 3 Report
Comments and Suggestions for Authors
Please see the attached document.

Author Response

(The authors gave the same response as above.)

Round 2
Reviewer 1 Report
Comments and Suggestions for Authors
The revised version is acceptable and ready for publication.
Author Response
Thank you once again for reviewing our revised manuscript. We are pleased to know that our responses have addressed your comments. We believe your suggestions have improved the quality of our manuscript.
Reviewer 2 Report
Comments and Suggestions for Authors
Thank you for correcting the manuscript. I accept all corrections and consider them comprehensive.
Author Response

(The authors gave the same response as above.)

Reviewer 3 Report
Comments and Suggestions for Authors
The authors have addressed my comments effectively and I endorse the paper. Two extremely minor typos:
- There is a missing "s" on line 65 (currently says "the amendment include") - I believe there's also a typo in line 277 but I may be misunderstanding the red correction font.Author Response
Thank you very much for reviewing our revised manuscript and your suggestion. We have incorporated your suggestions into the revised manuscript. We believe these edits have improved the quality of our manuscript. We have implemented all edits using the track change function, and a point-by-point response to each comment is provided for your review as shown in the following table.
|
There is a missing "s" on line 65 (currently says "the amendment include") |
Thank you for your valuable feedback and the time you have dedicated to reviewing our manuscript. We have made corrections to the grammatical errors identified, and we appreciate your suggestions once again. The detailed changes are as follows. (Page 2, line 50)
|
|
I believe there’s also a typo in line 277 but I may be misunderstanding the red correction font. |
Thank you very much for carefully reviewing our manuscript and pointing out spelling errors. We have rechecked the sentence in line 277 and made it more complete. The revised content is as follows. (Page 9, lines 252-253).
|

Round 3
Reviewer 3 Report
Comments and Suggestions for Authors
Looks great, very nice paper!